# Creep constitutive modeling of the shear strength of the permafrost-concrete interface considering the stress level at -1˚C

Fei He[1]*, Wanyu Lei[1], Erqing Mao[1], Qingquan Liu[1], Hangjie Chen[2], Xu Wang[3]

**1** School of Civil Engineering, Lanzhou Jiaotong University, Lanzhou, China, **2** Gansu Urban and Rural Planning Design and Research Institute Co. Ltd., Lanzhou, China, **3** National and Provincial Joint Engineering Laboratory of Road & Bridge Disaster Prevention and Control, Lanzhou, China

* hefei_2006@126.com

**Data Availability Statement:** We have uploaded the data into an online database, which can be found at the link below. https://doi.org/10.6084/m9.figshare.24935673.v2.

## Abstract

The shear creep characteristics of the contact surface between the permafrost and the structure play an important role in the study of the law of deformation and the measures for the prevention and control of pile foundations. In order to study the creep law and the development tendency of the contact surface between permafrost and concrete, it is necessary to establish an accurate creep model. In this study, based on the Nishihara model, a nonlinear element and damage factor D were introduced to establish an intrinsic model of permafrost-concrete contact surfaces considering the effect of shear stress. Creep tests with graded loading of concrete and frozen silt with different roughness at -1˚C were conducted using a large stress-controlled shear apparatus. The adequacy of the model was checked using the test data and the regularity of the parameters of the model was investigated. The results show that the creep curves of the contact surface obtained with the improved Nishihara model agree well with the test results and can better describe the whole process of creep of the contact surface of frozen concrete. The analysis of the experimental data shows that: the roughness of the concrete has an inhibiting effect on the creep deformation of the contact surface, When the roughness $R$ varies from 0 mm to 1.225 mm, the specimen corresponds to a long-term strength of 32.84 kPa to 34.57 kPa. For the same roughness and creep time, the creep deformation of the contact surface is more significant with the increasing shear stress $\tau$. The results of the study can provide a theoretical basis for the design and numerical simulation of pile foundations in permafrost regions.

## 1 Introduction

The issue of foundation settlement and deformation of cold region structures is a critical concern in the study of frozen soil engineering. It is essential for the normal functioning and safe service of the structure in permafrost regions. This problem has been exacerbated by the prolonged degradation of permafrost due to global climate change. Creep deformation in pile foundations is an important deformation mechanism during the operational phase of bridge piles in cold regions, and even relatively small loads can cause creep deformation in the piles

**Funding:** The research described in this paper was financially supported by the National Natural Science Foundation of China (Grant No. 41902272) and the Basic Research Innovation Group Project of Gansu Province, China (Grant No. 21JR7RA347).

**Competing interests:** The authors have declared that no competing interests exist.

[1]. Furthermore, frozen soil exhibits significant temperature sensitivity, adding complexity and instability to the creep behavior between piles and soil in cold regions. Consequently, there is an urgent need for research on the settlement and deformation patterns of pile foundations in frozen soil under different constant loads, as well as strategies for mitigating settlement. Research into shear creep tests at the frozen soil-structure interface and creep constitutive models plays a pivotal role in addressing such issues.

The creep deformation patterns at the frozen soil-structure interface are similar to those of frozen soil creep deformation and can be assessed through stress-controlled shear testing apparatus or triaxial testing equipment [2–5]. Creep deformation in frozen soil refers to the process of elastic-plastic-viscous deformation under constant loading conditions. It is primarily influenced by the temperature, moisture (ice) content, and stress conditions of the frozen soil. There are two types of permafrost foundation creep, attenuation and non-attenuation. The three broad categories of permafrost creep modeling techniques include empirical models, stress-strain-time models, and rheological models [6]. The most commonly employed approach in rheological modeling involves combining various mechanical elements with distinct characteristics to effectively describe the relationship between soil creep deformation and time [7]. Within rheological models, classical mechanical element models include the visco-elastic Kelvin body, the viscoplastic Bingham body, the Maxwell body, the Burgers model, and the Nishihara model, among others [8]. The Nishihara model can explain the attenuation and isotropic creep process in permafrost creep experiments better at lower loads [9]. But at higher stresses, the linear element in this model is unable to adequately capture the nonlinear rapid creep stage. Therefore, in order to more accurately describe the whole process of creep, both domestic and international scholars introduced the nonlinear rheological model [10].

Song et al. [11], and Sun et al. [12] pointed out that the creep viscosity coefficient η is a function of the applied load and the duration of the load. Deng et al. [13] proposed a non-Newtonian fluid viscous damper element and combined it with a traditional model to obtain a new rheomechanical model. Qi et al. [14] realized the description of accelerated creep stage and landslide near-slip prediction by connecting nonlinear dashpot pots with strain-triggered function in series on the basis of traditional Nishihara model. Hou et al. [15] introduced hardening parameters and damage variables in the Nishihara model to describe the creep process of frozen coarse-grained pulverized clay at different coarse-grained contents. Hou et al. [16] presented a nonlinear creep damage model for rocks considering initial damage by introducing an improved viscous unit and a new nonlinear viscoplastic damage unit. Li et al. [17] proposed an improved Nishihara model considering temperature and stress by introducing hardening factor and damage factor. Zhu et al. [18] modified the elastic modulus of the elastic component in the Nishihara model as a function of stress and introduced a damage variable to enhance the viscoplastic unit. This modification enables the model to accurately describe the entire creep process under various shear stress levels and temperatures. Xu et al. [19] introduced a nonlinear viscoplastic unit and proposed a creep model for describing the whole process of creep in kilomagnetite. The aforementioned enhancements to the Nishihara model primarily focus on a specific type of soil or rock. However, there is a relative dearth of research on the shear creep characteristics at the interface between frozen soil and concrete, particularly in the context of pile foundations in permafrost regions subjected to elevated temperatures due to permafrost degradation. Currently, the long-term deformation of pile foundations in permafrost regions primarily considers the structural deformation of the piles themselves, with limited attention given to the creep behavior at the interface between the pile and the surrounding soil.

Previous studies have shown that, Material surface roughness, soil density, moisture content, particle angle characteristics, structural material, and loading characteristics have

important effects on the shear behavior of the material-soil interface [20–22]. Kishida et al [23] concluded that roughness plays an important role in the friction between steel and soil surface. S Quanbin et al. [24] constructed an empirical formula on adfreezing strengths incorporating temperatures, normal stress, and roughness. They concluded that relationship between peak adfreezing strength and roughness satisfied a logarithmic function, The fluctuation cycle about strength curves of residual adfreezing strength increases with increasing roughness. Chen et al. [25] investigated different types of red clay-concrete interfaces using large-scale direct shear tests. The results showed that the peak shear strength values were close to the residual shear strength values at different normal stress levels. The shear strength of the clay-concrete interface is both cohesive and frictional, and the interfacial shear strength increases with the increase of surface roughness. Karam et al. [26] conducted a series of laboratory large-scale direct shear tests on sand-concrete specimens and sound that the shear strength characteristics of the sand-concrete interface are mainly affected by the relative density of sand, normal stress level, concrete surface roughness and interface area ratio ($Ar$). The increase of concrete surface roughness leads to the increase of mobilization friction between sand and concrete.

In summary, the aforementioned improvements to the Nishihara model have almost been proposed for specific soil or rock types. There is a relative lack of research on the shear creep characteristics of frozen soil-concrete contact surfaces, especially in the context of pile foundations in permafrost regions under high-temperature conditions resulting from permafrost degradation. Besides, the freezing strength test results of permafrost-structure contact surface obtained from the rapid shear test cannot be directly applied to the long-term deformation study of pile foundation, and cannot reveal the mechanism of the deterioration of the long-term bearing performance of pile foundation in the permafrost zone. This work is based on the Nishihara model and extends it by analogizing frozen soil creep behavior. The modification involves replacing the linear viscous Kelvin element with a nonlinear component, where the viscosity coefficient $\eta_1$ becomes a function of time and stress. Additionally, a damage variable $D$ is introduced within the viscoplastic Bingham model. These adaptations result in an improved Nishihara model that comprehensively accounts for time-dependent and stress-dependent behavior, specifically tailored for characterizing the creep phenomena at frozen soil-concrete interfaces. A large-scale stress-controlled shear instrument was used to carry out graded loading creep tests on concrete with different roughness and frozen soil at -1°C, and the rationality of the model was verified on the basis of the test data. Findings from current study can be useful in guiding the design and theoretical research of pile foundations in permafrost regions.

## 2 Creep model of permafrost-concrete contact surface

The Nishihara model, which uses Kelvin and Bingham bodies in succession to describe the creep curve of the permafrost-concrete contact surface (Eq (1)), and the mechanical model is shown in Fig 1.

$$\gamma = \begin{cases} \dfrac{\tau}{G_0} + \dfrac{\tau}{G_1}\left[1 - \exp\left(-\dfrac{G_1}{\eta_1}t\right)\right], \tau < \tau_s \\[3ex] \dfrac{\tau}{G_0} + \dfrac{\tau}{G_1}\left[1 - \exp\left(-\dfrac{G_1}{\eta_1}t\right)\right] + \dfrac{\tau - \tau_s}{\eta_2}t, \tau \geq \tau_s \end{cases} \tag{1}$$

Where $G_0$ and $G_1$ are the Hooke body shear modulus and Kelvin body shear modulus, respectively. $\eta_1$ and $\eta_2$ are the Kelvin body viscosity coefficient and viscoplastic body viscosity

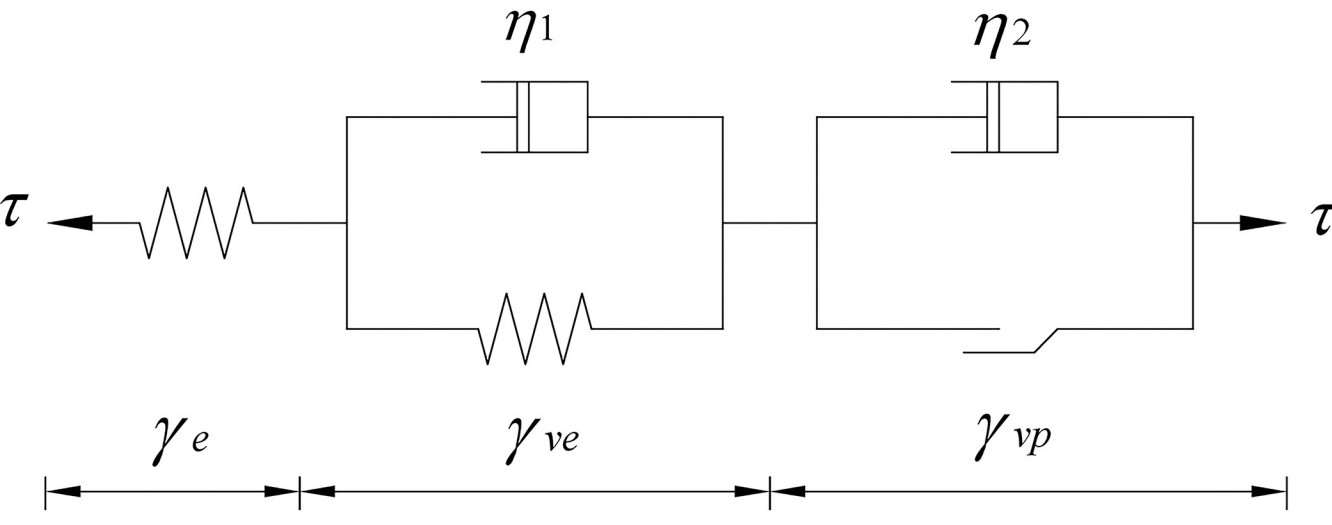

**Fig 1. Nishihara model.**

coefficient, respectively. $\tau$ is the creep stress, $\tau_s$ is the long-term strength limit and $\gamma$ is strain value.

Since the Nishihara model cannot describe the accelerated phase of shear creep, its mechanical components were modified to obtain the improved Nishihara model, which is depicted mechanically in Fig 2. The following improvements are listed.

### 2.1 Viscoelastic Kelvin body

The viscoelastic Kelvin model is obtained by connecting a Hooke elastic body with a Newton viscous body and is capable of describing the nonlinear creep decay phase of the frozen soil-concrete. According to the analysis of frozen Qinghai-Tibetan chalk-concrete contact surface test results, Creep has non-linear characteristics, the degree of nonlinearity is related to the creep time, and the creep process of the viscous coefficient of hardening with time increases

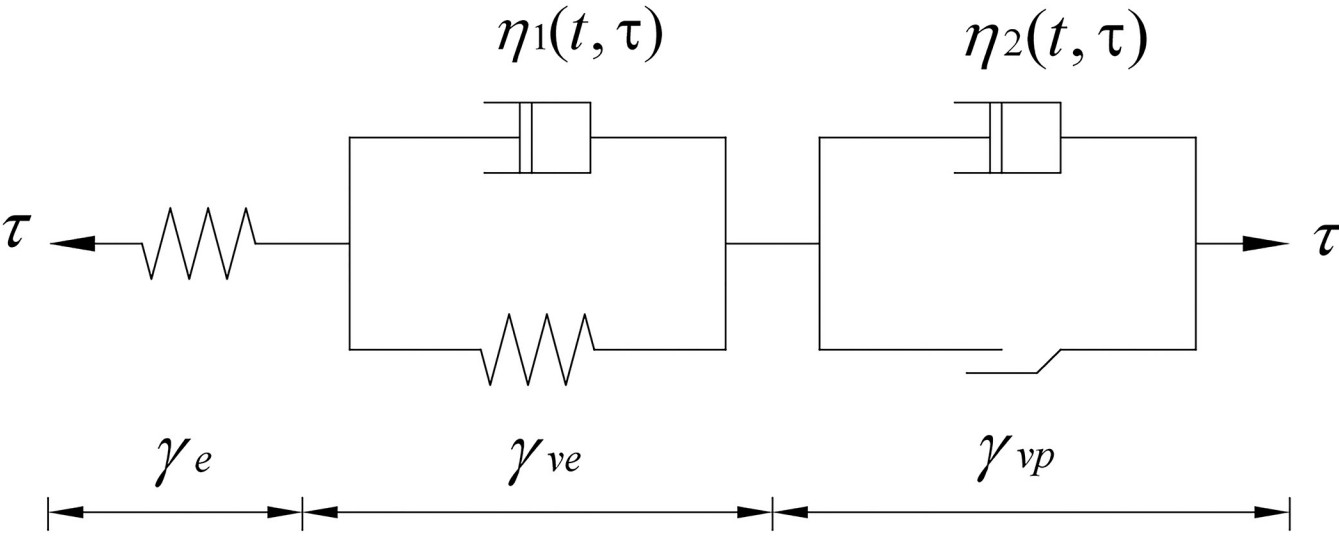

**Fig 2. Modified Nishihara model.**

the law of increasing. Assuming that the viscosity coefficient $\eta_1$ is a function of stress level and time [27, 28], it can be expressed as shown in Eq (2):

$$\eta_1(t, \tau) = \eta_1^0 A(\tau) \left( \frac{t^{1/2}}{t^{1/2} + 0.1} \right) \tag{2}$$

Where $\eta_1^0$ represents the initial viscosity coefficient of the Kelvin model; $\tau$ denotes the shear stress level, and $t$ signifies the creep time.

Considering that with the increase of shear stress, the longer the time for the specimen to reach the steady state creep stage, so the Kelvin body viscous coefficient will increase with the increase of shear stress, combined with the previous study, it is assumed that the relationship between the effect of shear stress level on the viscous coefficient of the Kelvin body is shown in Eq (3) [29, 30]:

$$A(\tau) = \tau^\beta \tag{3}$$

Eq (2) is derived for t as shown in Eq (4):

$$\dot{\eta_1}(t, \tau) = \eta_1^0 A(\tau) \frac{0.1 \cdot t^{-1/2}}{2(t^{1/2} + 0.1)^2} > 0 \tag{4}$$

In Eq (4), $\cdot \eta_1(t, \tau) > 0$, which indicates that the viscous coefficient with the increase in time shows a monotonous increasing trend, in line with its rule of change. From Eq (2), it can be seen that if $t = 0$, $\eta_1(t,\tau) = 0$, if $t \to \infty$, $\eta_1(t, \tau) = \eta_1^0 A(\tau)$, which show that with the continuous growth of creep time, the improved Kelvin body hysteresis coefficient monotonically increasing from 0 to $\eta_1^0 A(\tau)$. Since an increase in the hysteresis coefficient will prevent the shear rate from growing, it can be utilized to characterize the lower shear stress under a reduced creep stage. The improved Kelvin body constitutive equation is shown in Eq (5):

$$\tau = G_1 \gamma_{ve}(t) + \eta_1(t, \tau) \frac{d\gamma_{ve}(t)}{dt} \tag{5}$$

Where $G_1$ is the Kelvin body shear modulus and $\eta_1$ is the Kelvin body viscosity coefficient. $\gamma_{ve}$ is the viscoelastic strain.

To get the Kelvin body shear creep equation illustrated in Eq (6), substitute Eq (2) into Eq (5), using $t = 0$ and $\gamma_{ve}(t) = 0$ as the initial value:

$$\gamma_{ve}(t) = \frac{\tau}{G_1} \left[ 1 - \exp\left( -\frac{G_1}{\eta_1^0 A(\tau)} \left( t + 0.2 t^{1/2} \right) \right) \right] \tag{6}$$

This part is used to describe the deformation characteristics of the frozen soil-concrete contact surface during the decay creep stage.

## 2.2 Viscoplastic Bingham body

Вялов introduced the concept of "damage" into the study of permafrost mechanics for the first time [31, 32]. Damage and viscoplastic flow are the causes of the material's energy consumption under specific circumstances [33]. Damage mechanics is more suited to understanding the deformation and damage of permafrost because it can explain the full process of structural destruction. In order to characterize the developmental (accelerated) creep phase, the damage variable $D$ is included. The Weibull distribution is used to define the damage variable $D$ as

shown in Eq (7) and the viscoplastic strain as shown in Eq (8) [34, 35]:

$$D(t, \tau) = \begin{cases} 0, \tau \leq \tau_{\mathrm{u}} \\ \\ 1 - \exp(-nt^m), \tau > \tau_{\mathrm{u}} \end{cases} \tag{7}$$

$$\gamma_{vp} = \begin{cases} 0, \tau < \tau_{\mathrm{s}} \\ \dfrac{\tau - \tau_{\mathrm{s}}}{\eta_2(t, \tau)} t, \tau \geq \tau_{\mathrm{s}} \end{cases} \tag{8}$$

$$\eta_2(t, \tau) = \eta_2^0 (1 - D) = \begin{cases} \eta_2^0, \tau \leq \tau_{\mathrm{u}} \\ \eta_2^0 \exp(-nt^m), \tau > \tau_{\mathrm{u}} \end{cases} \tag{9}$$

Where $\eta_2^0$ is the initial viscous coefficient of Bingham body, $\tau_s$ is the long-term strength limit and $\tau_u$ is the long-term strength, $n$, $m$ are the damage factor parameters, which can be obtained by the least squares method.

The improved model is composed of Hooke elastomer, Kelvin body and Bingham body in series, and the total strain is the sum of the strains in each part, which is obtained by substituting into Eq (3) and Eq (9):

$$\gamma = \gamma_e + \gamma_{ve} + \gamma_{vp} = \begin{cases} \dfrac{\tau}{G_0} + \dfrac{\tau}{G_1}\left[1 - \exp\left(-\dfrac{G_1}{\eta_1^0 \tau^\beta}\left(t + 0.2t^{1/2}\right)\right)\right] \\ , \tau < \tau_{\mathrm{s}} \\ \dfrac{\tau}{G_0} + \dfrac{\tau}{G_1}\left[1 - \exp\left(-\dfrac{G_1}{\eta_1^0 \tau^\beta}\left(t + 0.2t^{1/2}\right)\right)\right] \\ + \dfrac{\tau - \tau_{\mathrm{s}}}{\eta_2^0} t, \tau_{\mathrm{s}} \leq \tau \leq \tau_{\mathrm{u}} \\ \dfrac{\tau}{G_0} + \dfrac{\tau}{G_1}\left[1 - \exp\left(-\dfrac{G_1}{\eta_1^0 \tau^\beta}\left(t + 0.2t^{1/2}\right)\right)\right] \\ + \dfrac{\tau - \tau_{\mathrm{s}}}{\eta_2^0 \exp(-nt^m)} t, \tau > \tau_{\mathrm{u}} \end{cases} \tag{10}$$

Eq (10) represents the improved constitutive model for the creep behavior of the frozen soil-concrete interface. Its applicability has been validated through a series of creep tests on the frozen silt-concrete interface using specific conditions (as described in Section 3) with a moisture content of 22%, normal stress of 150 kPa, and a roughness parameter of $R = 0.538$ mm. The creep curves under various shear stress levels were individually fitted to the experimental data. The fitting results are illustrated in Fig 3, and the coefficients of determination are presented in Table 1. The correlation coefficients squared values exceed 0.99, demonstrating that this model effectively characterizes the creep behavior of the frozen soil-concrete interface under different shear stress levels.

In order to unify the factor of shear stress level and establish the contact surface creep model considering the shear stress level, according to the above conclusions and synthesize the results of the frozen silt-concrete contact surface creep test, the relationship between $G_0$, $G_1$, $\eta_1^0$ and the shear stress $\tau$ can be defined as an exponential function form as shown in Eq (11) ~

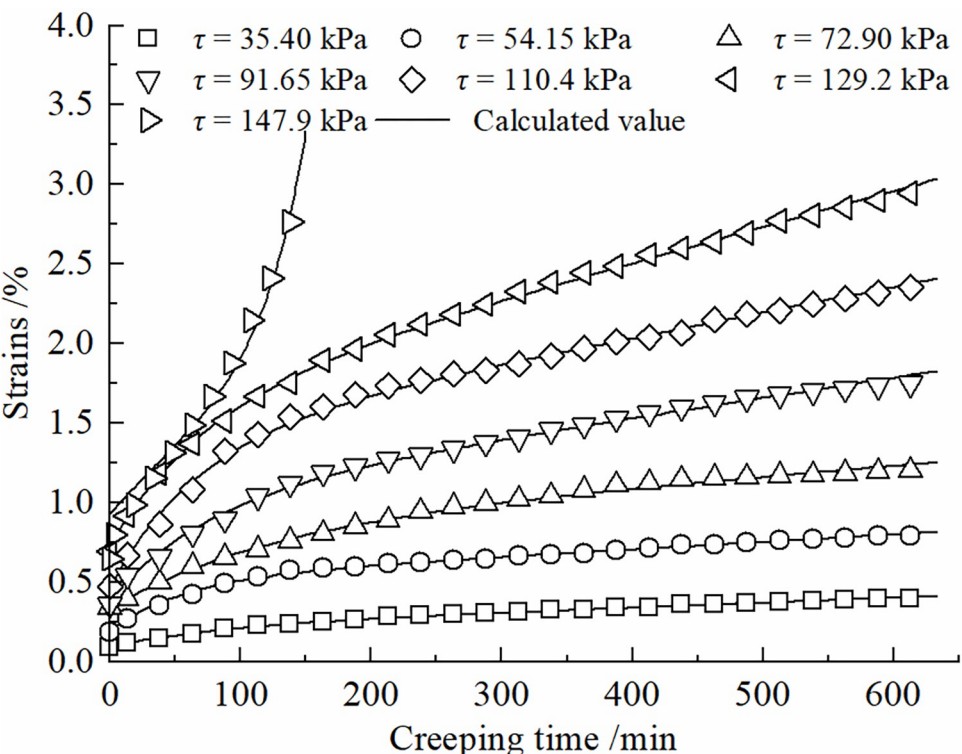

**Fig 3. Validation of the applicability of creep constitutive model for the interface of frozen soil and concrete.**

(14) respectively:

$$G_0(\tau) = a\exp\left(-\frac{\tau}{b}\right) + c \tag{11}$$

$$G_1(\tau) = d\exp\left(-\frac{\tau}{e}\right) + f \tag{12}$$

$$\eta_1^0 = g\exp\left(-\frac{\tau}{h}\right) + i \tag{13}$$

Where $a$, $b$, $c$, $d$, $e$, $f$, $g$, $h$ and $i$ are model parameters, $\tau$ is the shear stress. $\eta_2^0$ firstly increases, then decreases, and is expressed as a quadratic function:

$$\eta_2^0 = j\tau^2 + k\tau + l \tag{14}$$

Where $j$, $k$, $l$ are model parameters and $\tau$ is shear stress.

**Table 1. Evaluation of model goodness of fit.**

| $\tau$ / kPa | 35.40 | 54.15 | 72.90 | 91.65 | 110.40 | 129.15 | 147.9 |
|---|---|---|---|---|---|---|---|
| Root mean quare error | 5.8572E-5 | 8.2741E-5 | 1.9865E-4 | 2.2299E-4 | 2.7839E-4 | 2.0168E-4 | 1.1420E-3 |
| Sum of squared residuals | 4.4598E-5 | 8.8999E-5 | 5.1298E-6 | 6.87314E-6 | 1.0075E-5 | 5.2875E-6 | 4.3037E-5 |
| Square of correlation coefficient | 0.995186 | 0.996916 | 0.993603 | 0.996169 | 0.996716 | 0.998902 | 0.976818 |
| Coefficient of determination | 0.994977 | 0.996841 | 0.993542 | 0.996033 | 0.99665 | 0.998844 | 0.970974 |
| Chi-square | 8.6167E-5 | 9.2918E-5 | 2.4834E-4 | 2.9361E-4 | 4.6687E-4 | 1.2210E-4 | 1.9396E—3 |
| F-statistic | 26462.91 | 41371.70 | 19880.11 | 33282.96 | 38850.38 | 116473.13 | 1306.25 |

The Eq (10)–Eq (14) above illustrate the improved creep intrinsic model of the permafrost-concrete contact surface taking the stress level into account.

## 3 Experimental procedure

### 3.1 Testing apparatus

Creep Shear Instrument. The aforementioned improved Nishihara model was validated using experimental results conducted on the frozen silt-concrete interface. The testing apparatus employed was a large-scale stress-controlled shear device developed in-house at the Geotechnical Laboratory of Lanzhou Jiaotong University (as depicted in Fig 4), which mainly consists of three parts: the main frame, the shear box and the stress loading system. The dimensions of both the upper and lower shear boxes were 200×200×100 mm³ (length × width × height). The normal pressure was applied by a hydraulic jack through a reaction beam, while horizontal shear stress was incrementally applied using 25 kg bags of iron sand through pulleys.

Refrigeration and temperature control systems. XT5701LTB-450 high and low temperature cold bath system was selected to control the temperature of the cryogenic chamber, which can realize the temperature control accuracy of ± 0.1°C. Pt-100 Platinum resistance sensor was used for temperature measurement.

### 3.2 Specimen production

The concrete portion of the shear specimen was pre-fabricated with vertical grooves on its surface to create different roughness levels. The mix proportions of concrete are detailed in Table 2. In the experiments, four different concrete specimens with distinct surface roughness were designed, and their detailed geometric dimensions are illustrated in Fig 5. Surface roughness for these four concrete structures was measured using the sand-pouring method, resulting in values of 0 mm, 0.538 mm, 0.775 mm, and 1.225 mm, respectively. The principle behind this method involves using the volume replacement technique with standard sand to measure the volume of depressions within the concrete surface area. This measurement provides the average depth of these depressions, which serves as a quantification of the concrete surface roughness, as shown in Eq (15):

$$R = \frac{V_s}{A} \tag{15}$$

Where $R$ is the concrete surface roughness; $V_s$ is the volume of standard sand in the groove; $A$ is the surface area of the contact surface.

The test soil is Lanzhou silt, and the physical properties of the soil samples are shown in Table 3. The particle size distribution of silt is shown in Table 4, and the moisture content of the soil samples is 22%. Inside the sample box with the precast concrete structure at the bottom, the silt was layered and compacted to form an integrated silt-concrete structure, with temperature sensors (Pt100) embedded at the center and edges of the contact surface, as well as at the height center of the soil sample.

### 3.3 Test method

After sample preparation, the entire assembly, along with the sample, was rapidly frozen at -20°C for 24 hours in a low-temperature chamber. Subsequently, it was subjected to constant temperature and pressure treatment in a low-temperature test chamber according to the target conditions (temperature of -1°C, normal pressure of 150 kPa). When conditions stabilized, the horizontal shear stress was applied in a graded manner. Creep loading conditions are shown

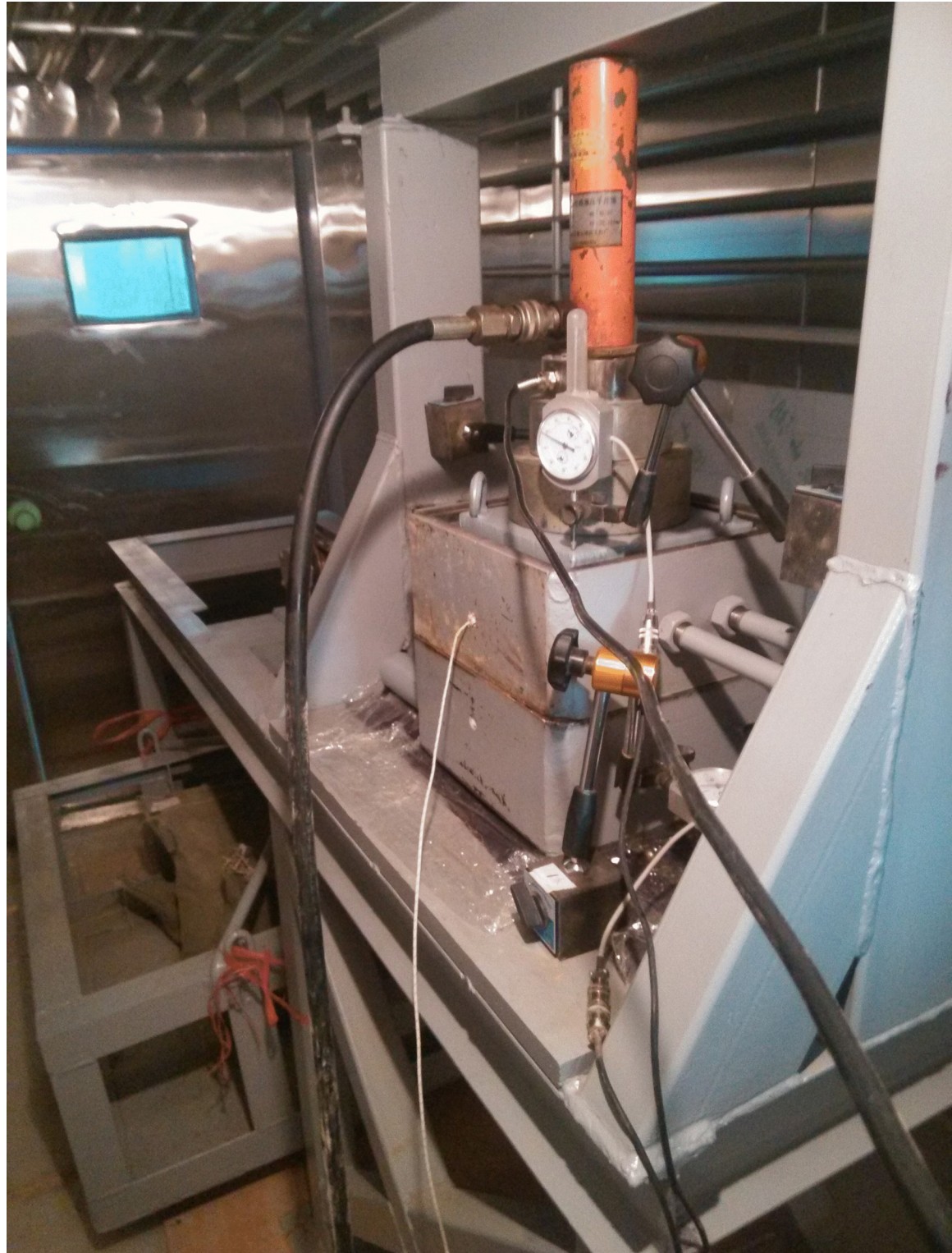

**Fig 4. Large-scale stress-controlled shearing apparatus.**

**Table 2. Mix proportion of concrete (unit: kg /m3).**

| Cement | Sand | Gravel | Water |
|--------|------|--------|-------|
| 445 | 606 | 1125 | 195 |

in Table 5. The entire shear process took place within the low-temperature test chamber, with shear displacement data automatically collected using an electronic dial gauge connected to a computer. The creep data obtained during the hierarchical loading process couldn't directly describe the shear creep behavior of the frozen silt-concrete interface under individual shear stress levels. Therefore, the "Chen's deformation superposition method" was employed to transform the creep test data from the entire loading process into creep data corresponding to different individual shear stress levels [36].

In this creep test, the stability criteria are as follows: during the creep decay phase, the strain rate is less than or equal to 0.001 mm/h; in the steady-state creep phase, the shear creep strain rate reaches a stable value; and during the accelerated creep phase, specimen failure occurs.

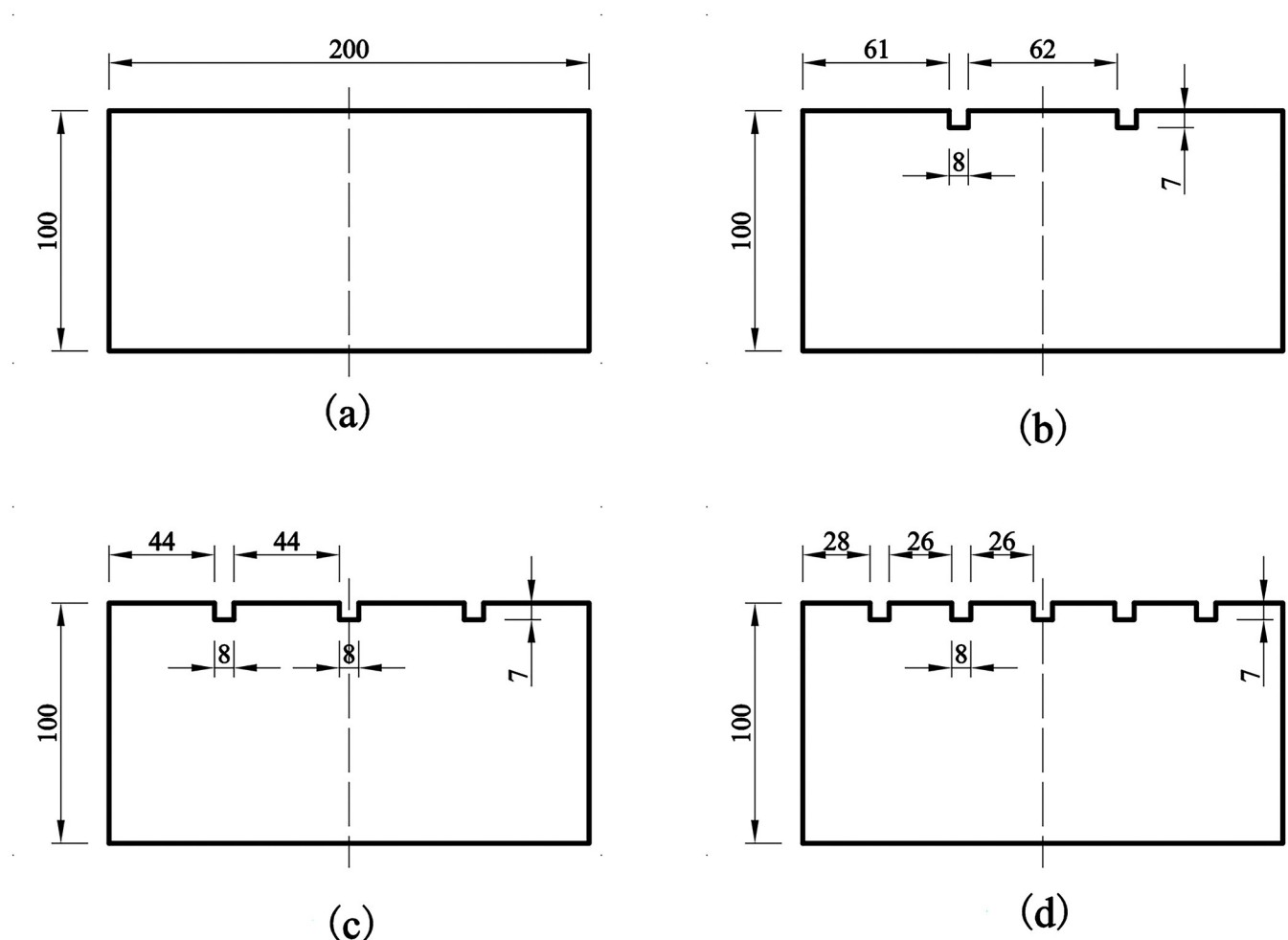

**Fig 5. Surface roughness design of concrete test block (unit: mm).** (a) $R = 0$ mm; (b) $R = 0.538$ mm; (c) $R = 0.775$ mm; (d) $R = 1.225$ mm.

**Table 3. Silt physical index.**

| Optimum water content/% | Maximum dry Density/(g·cm⁻³) | Liquid Limit Content water content /% | Plastic Limit Water content /% | plasticity index |
|---|---|---|---|---|
| 17.79 | 1.73 | 27.1 | 17.4 | 9.7 |

**Table 4. Particle size distribution of silt.**

| Type of soil | Particle sizes distribution of different particle size (%) | | | |
|---|---|---|---|---|
| | > 0.1 mm | 0.1 ~ 0.05 mm | 0.05 ~ 0.005 mm | < 0.005 mm |
| Silt | 6.1 | 24.6 | 62.3 | 7.0 |

**Table 5. Creep test conditions.**

| Roughness / mm | Sheer stress / kPa |
|---|---|
| 0 | 35.4→54.15→72.9→91.65→110.4 |
| 0.538 | 35.4→54.15→72.9→91.65→110.4→129.15→147.9 |
| 0.775 | 35.4→54.15→72.9→91.65→110.4→129.15→147.9→166.65 |
| 1.225 | 35.4→54.15→72.9→91.65→110.4→129.15→147.9→166.65→185.4→204.15 |

# 4 Result and discussion

## 4.1 Long-term strength

Based on the observed patterns in the experimental data, the strain value corresponding to the maximum damage rate during the accelerated phase is defined as the strain at specimen failure. The time corresponding to the maximum damage rate is determined by Eq (16), and this time can be used to calculate the strain at failure:

$$[1 - \exp(-nt_\mathrm{f}{}^m)]'' = 0 \tag{16}$$

then

$$nm\exp(-nt_\mathrm{f}{}^m)[(m-1)t_\mathrm{f}{}^{m-2} - nmt_\mathrm{f}{}^{2(m-1)}] = 0 \tag{17}$$

Where $n$, $m$ are the damage factor parameters, and $t_\mathrm{f}$ is the damage time of the specimen.

According to the above discriminating criteria, the curve of the long-term intensity is determined by fitting the first-order decay exponential function as shown in Eq (18):

$$\tau_\mathrm{f} = B\exp\left(-\frac{t_\mathrm{f}}{C}\right) + \tau_\mathrm{s} \tag{18}$$

Where $\tau_\mathrm{f}$ is the shear stress causing damage at time $t_\mathrm{f}$; $\tau_\mathrm{s}$ is the ultimate long-term strength;4, $t_\mathrm{f} \to \infty$, $\tau_f = \tau_s$; $B$ and $C$ are ultimate long-term strength parameters.

Assuming that the shear stress is less than the shear stress associated with the accelerated creep phase, steady-state creep occurs. Based on the resulting failure strain values, long-term strength curves under different conditions were plotted, as depicted in Fig 6, with relevant parameters detailed in Table 6. The ultimate long-term strength $\tau_\mathrm{s}$ for specimens with roughness values of 0 mm, 0.538 mm, 0.775 mm, and 1.225 mm were found to be 32.84 kPa, 33.45 kPa, 33.59 kPa, and 34.57 kPa, respectively.

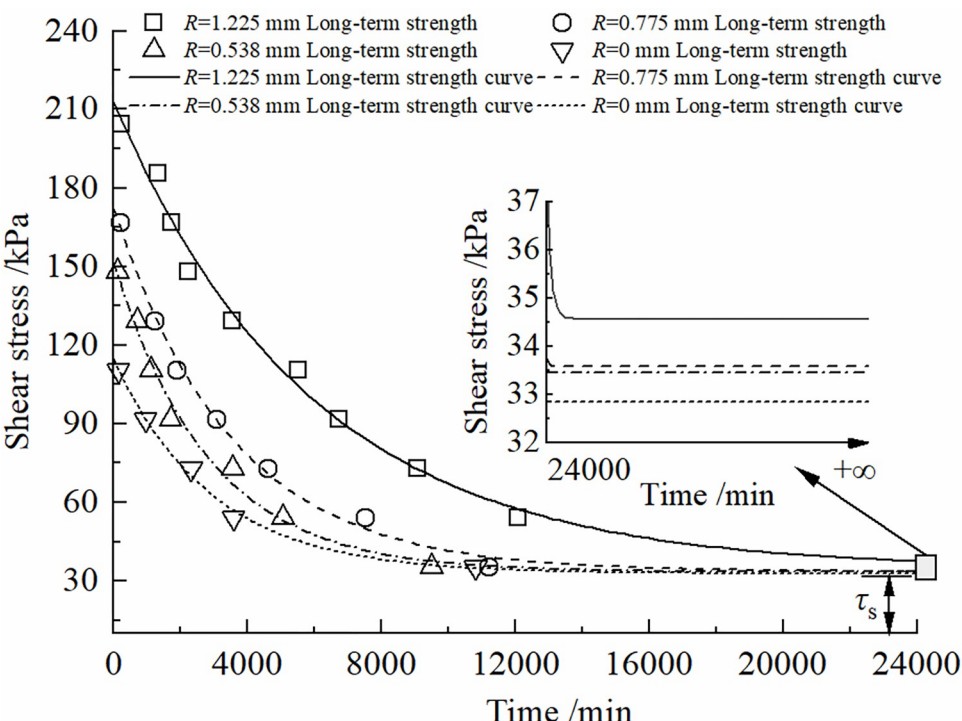

**Fig 6. Long-term strength curve of frozen soil-concrete interface under different conditions.**

## 4.2 Model verification

The degree of nonlinear creep of the contact surface is related to the creep time, and the viscous coefficient in the creep process shows a hardening increase law with time, and the increase of viscous coefficient will inhibit the growth of shear rate. The Kelvin body shear creep equation of Eq (6) can be used to describe the decaying creep stage at lower shear stresses. The introduction of the damage variable $D$ can be used to describe the developmental (accelerated) creep stage. Therefore, the improved model can more accurately simulate the creep curves of the frozen silt -concrete surface under different shear stress levels. Fig 7 presents a comparison between the shear creep test results of the frozen silt-concrete interface and the calculated values obtained using the improved Nishihara model. In this study, different roughness was formed by setting vertical grooves. The roughness of the experimental setup has a certain gap with the actual engineering, and the test results will have a certain difference. The data in Figs 7–10 show that when the roughness $R$ is 1.225 mm, the difference between the experimental data and the simulation results is larger compared to the other roughness, Therefore, the effect of roughness on the creep modeling of permafrost-concrete contact surfaces is yet to be thoroughly investigated.

**Table 6. Long-term strength curve parameters under different conditions.**

| Types of specimens | $B$ / kPa | $C$ / kPa | $\tau_S$/ kPa |
|---|---|---|---|
| $R$ = 0 mm | 81.73404 | 2937.62367 | 32.84121 |
| $R$ = 0.538 mm | 118.50757 | 2821.31829 | 33.45004 |
| $R$ = 0.775 mm | 138.76934 | 3508.00902 | 33.58962 |
| $R$ = 1.225 mm | 178.38600 | 5879.20963 | 34.57338 |

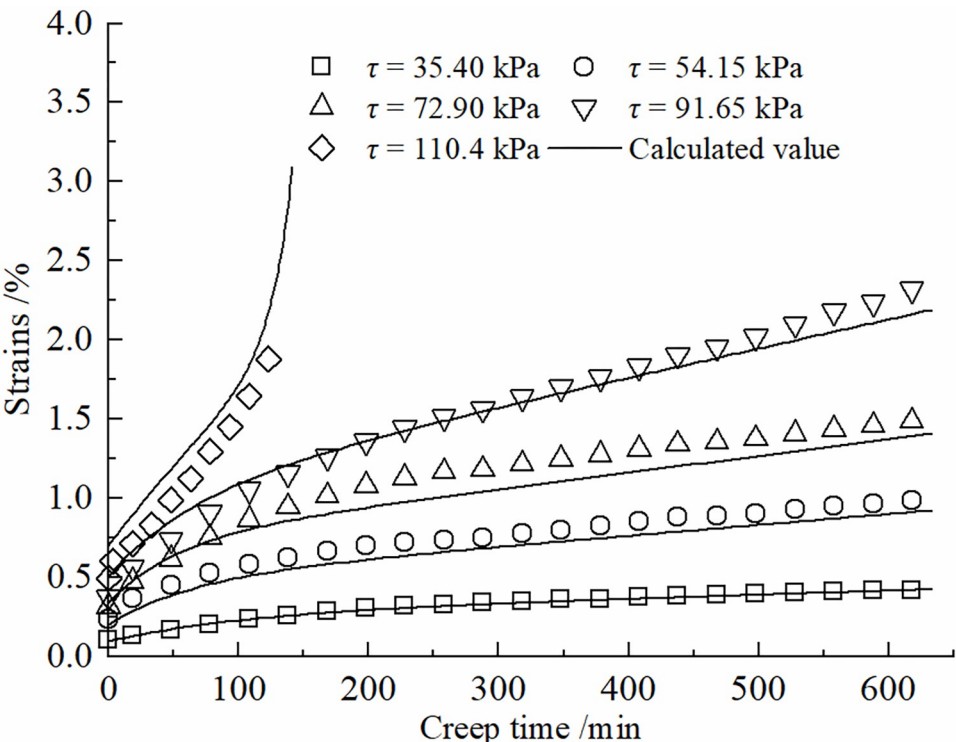

**Fig 7. Comparison of experimental and theoretical values of the interface at R = 0 mm.**

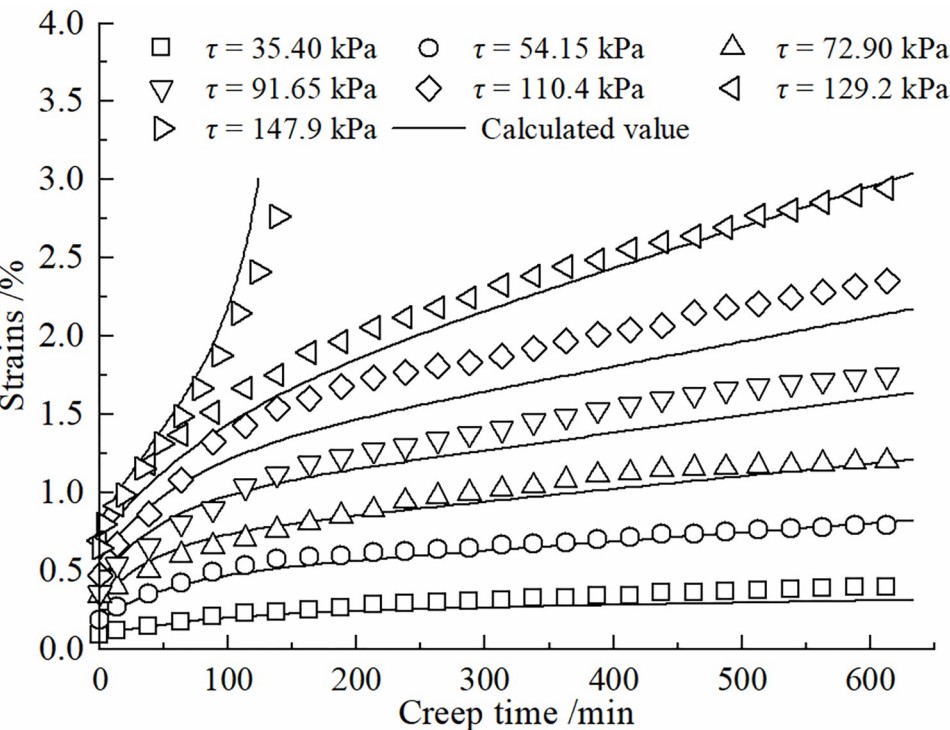

**Fig 8. Comparison of experimental and theoretical values of the interface at R = 0.538 mm.**

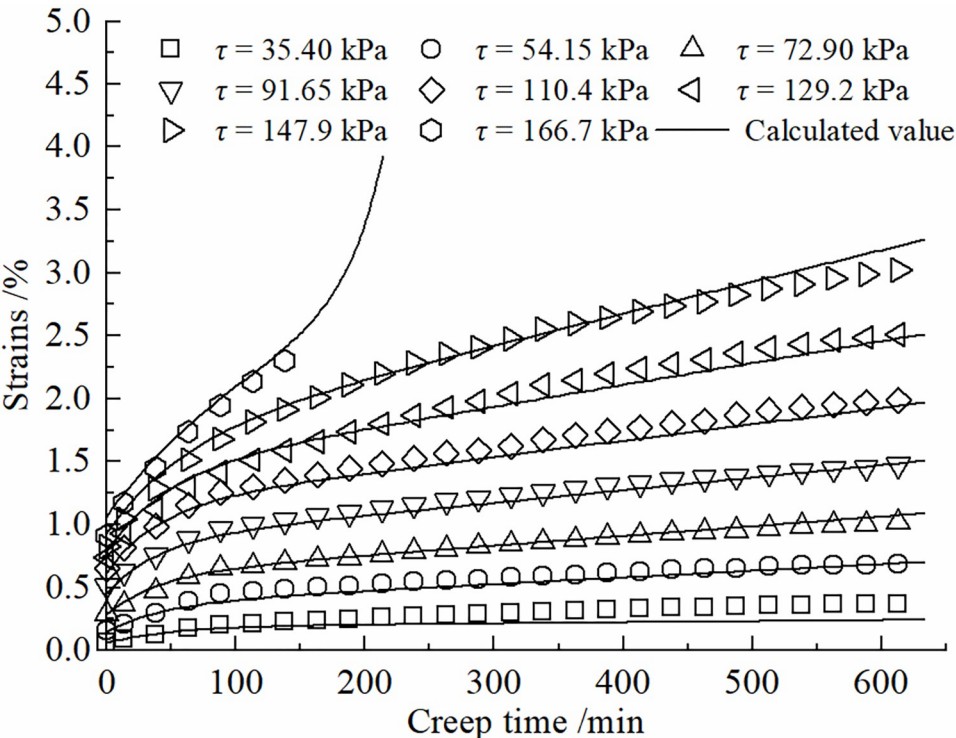

**Fig 9. Comparison of experimental and theoretical values of the interface at R = 0.775mm.**

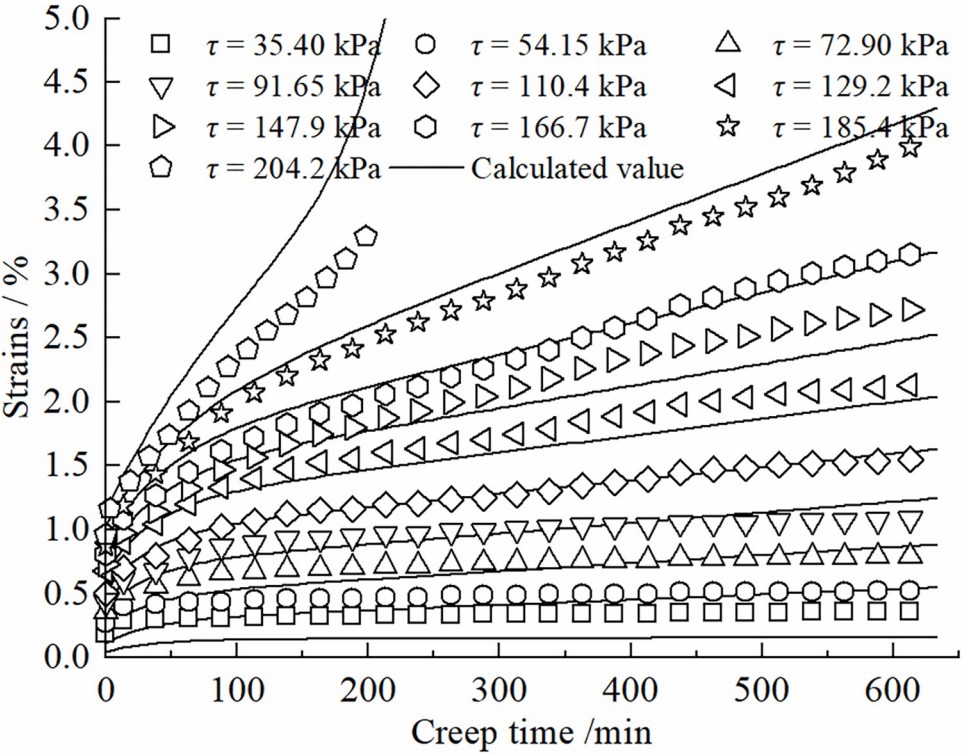

**Fig 10. Comparison of experimental and theoretical values of the interface at R = 1.225mm.**

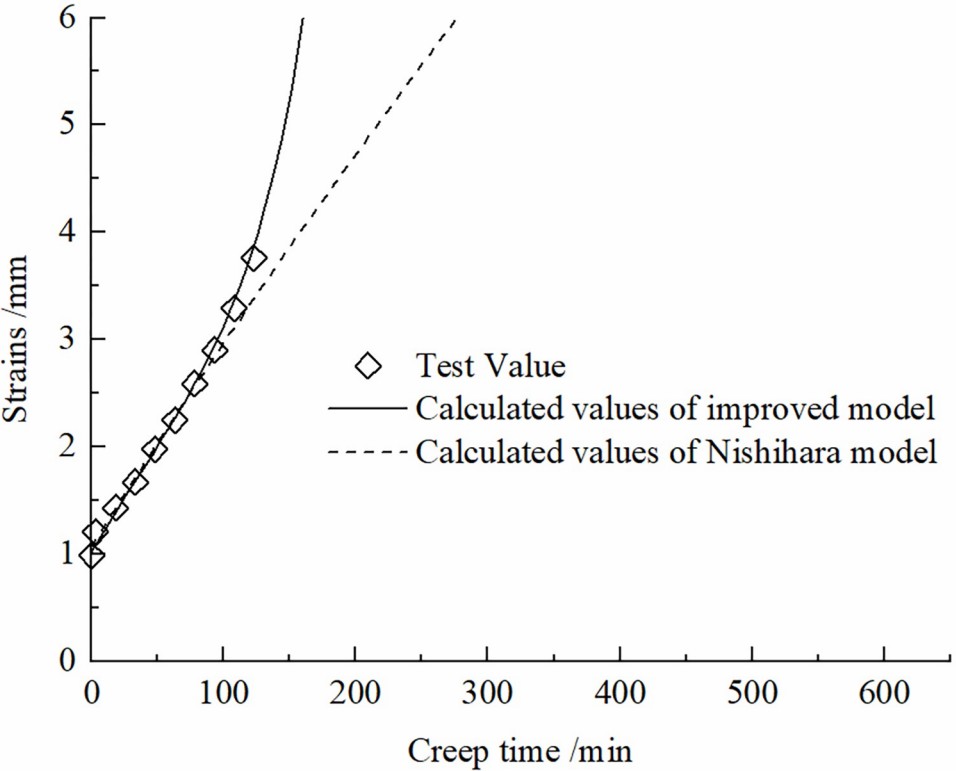

**Fig 11. Comparison of effect before and after model improvement.**

Fig 11 illustrates the conformity of the calculated values from the improved model and the traditional Nishihara model with the experimental data. It is evident that the improved model provides a better description of the shear creep test results at the accelerated phase of the frozen silt-concrete interface.

## 4.3 Parameters analysis

The relevant parameters in the improved model can be obtained from the physical interpretation and mathematical relationships within the creep curves. These model-related parameters are presented in Table 7, determined through the application of the Levenberg-Marquardt nonlinear least-squares optimization algorithm for fitting.

The effects of roughness $R$ and shear stress $\tau$ on the parameters of the improved Nishihara model are shown in Figs 12–15. It can be concluded that the shear elastic modulus $G_0$ of the Hooke body, the shear elastic modulus $G_1$ of the Kelvin body, and the viscous hysteresis coefficient $\eta_2^0$ of the Bingham body in the improved model grow as the increase of the roughness. However, the reverse is true for the variation of $\eta_1^0 \tau^\beta$ in Kelvin bodies; With the increase of shear stress, $G_0$ and $G_1$ demonstrate a tendency to decline and tend toward a constant value; $\eta_1^0 \tau^\beta$ decreases and then increases, whereas $\eta_2^0$ increases and then decreases.

Pile foundations in permafrost regions are usually located deep in the foundation, and the contact surface between concrete and soil may contain different soil layers, and the physical and mechanical properties of different soils may also have an effect on the creep characteristics of the contact surface. The experimental apparatus and numerical simulations also differ in the setting of roughness from the actual engineering, which has an effect on the creep

**Table 7. Long-term strength curve parameters under different conditions.**

| Roughness of specimen contact surface (unit: mm) | $G_0$ / kPa | | | $G_1$ / kPa | | |
|---|---|---|---|---|---|---|
| | a / kPa | b / kPa | c / kPa | d / kPa | e / kPa | f / kPa |
| R = 0 | 55646.56779 | 41.90391 | 11676.28003 | 18687.85536 | 19.70397 | 17549.35510 |
| R = 0.538 | 64935.37602 | 30.47369 | 17343.09271 | 72720.25583 | 15.37829 | 18239.11377 |
| R = 0.775 | 169206.16190 | 25.98796 | 15370.98316 | 17755.05903 | 48.37432 | 19022.89578 |
| R = 1.225 | 253067.80818 | 25.35250 | 17683.80853 | 29737.07203 | 46.75109 | 22144.22834 |

| $\eta_1$/ kPa·min | | | | $\eta_2$/ kPa·min | | | | |
|---|---|---|---|---|---|---|---|---|
| g | h | i | β | j | k | l | n | m |
| kPa min | kPa min | kPa min | | Min kPa$^{-1}$ | min | kPa min | | |
| 1397158.9927.0 | -14.1701 | 554603072.119 | -1.5620 | -1905.0527 | 280395.12303 | -6.494E6 | 3300752872588 | 5.7814 |
| 122588422889 | -20.9969 | 88383628482.5 | -3.5727 | -1164.5581 | 215643.31677 | -4.708E6 | 30002432.0955 | 3.5030 |
| 54905.13034 | -27.2881 | 11204741.3133 | -0.5932 | -785.08197 | 165847.17196 | -2.808E6 | 600327179477 | 5.0133 |
| 213355.24437 | -47.4024 | 6492475.2367 | -0.4848 | -690.35444 | 159332.13948 | -1.899E6 | 1306012129221 | 5.0513 |

characteristics of the shear surface. These limitations were not considered in this study. The long-term stability of pile foundations is important for the safe service of bridges in permafrost regions. Foundations in frozen soil regions exhibit significant viscoelastic behavior. Currently, the primary means of studying this behavior involve field monitoring and model testing. By defining the contact relationship using a creep constitutive model for the frozen soil-concrete interface, it is possible to numerically simulate the rheological properties of foundations in frozen soil areas. This approach offers a solution to some of the limitations that other methods face in the temporal dimension. It is instructive for the study of long-term creep deformation of pile foundations in permafrost regions.

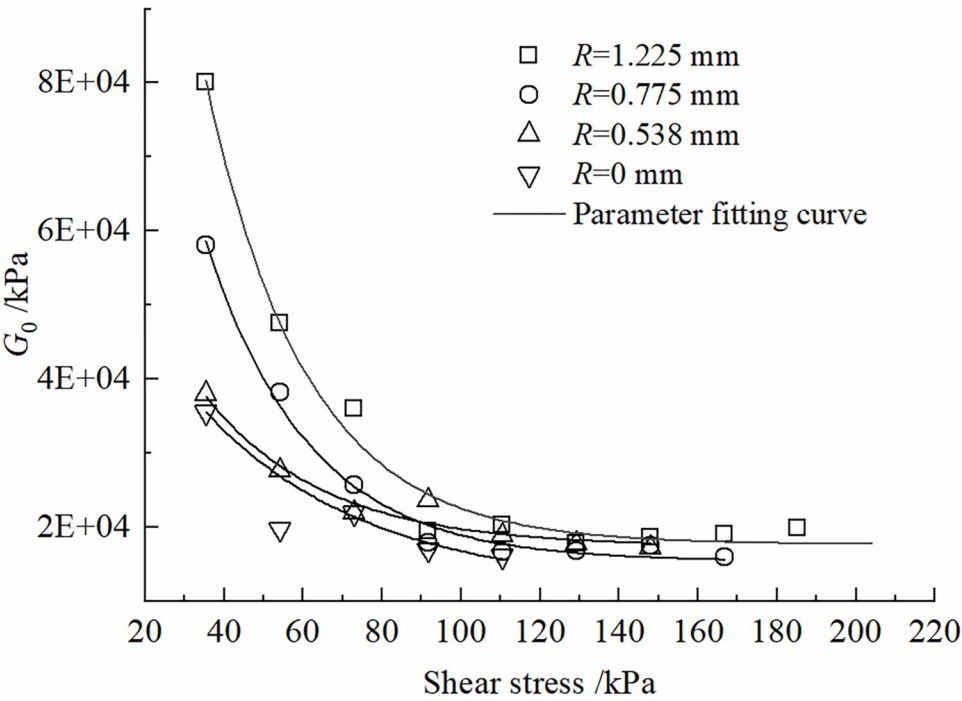

**Fig 12. Variation of $G_0$ with shear stress change rule.**

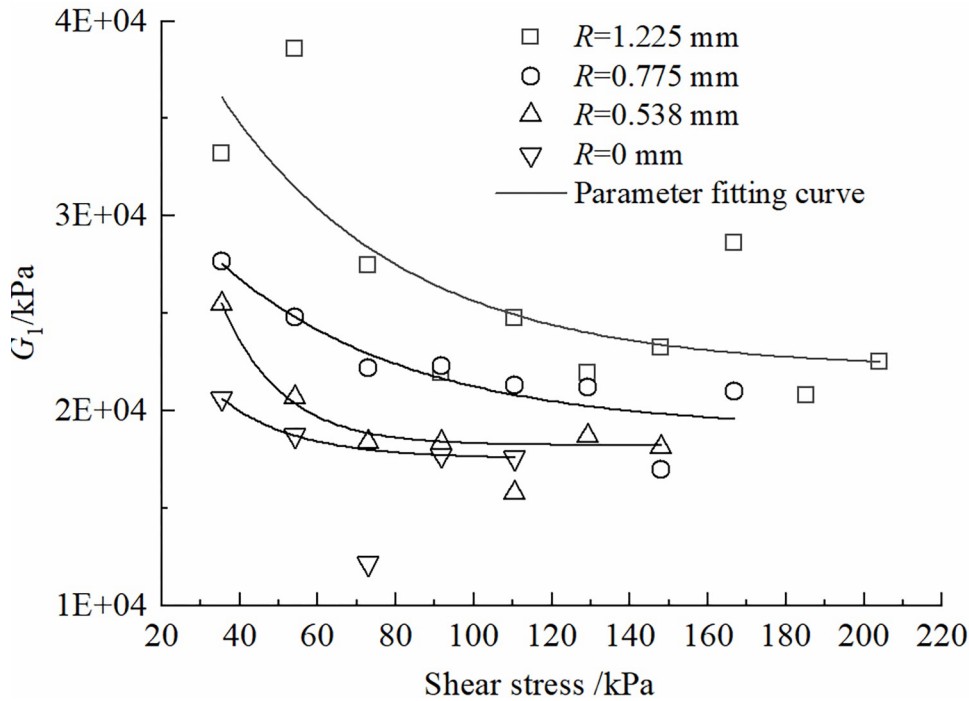

**Fig 13. Variation of $G_1$ with shear stress change rule.**

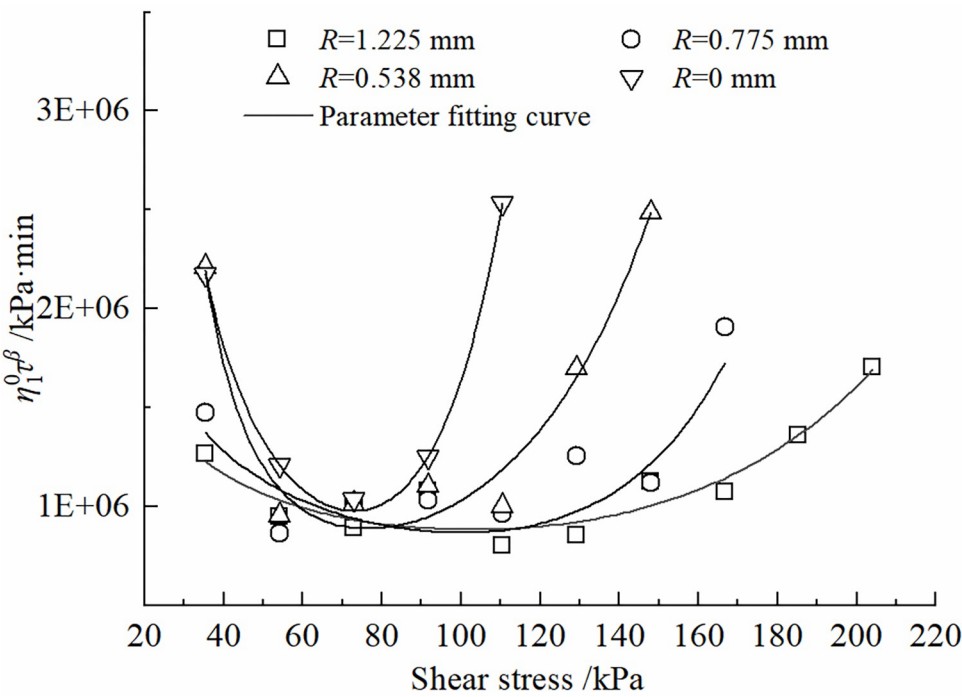

**Fig 14. Variation of $\eta_1^0 \tau^\beta$ with shear stress change rule.**

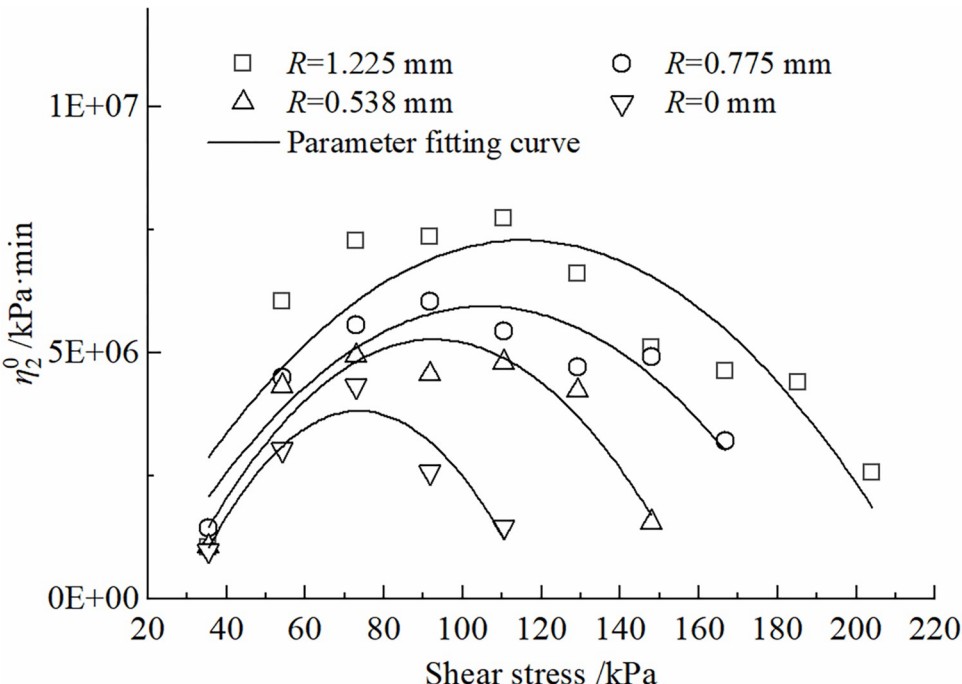

**Fig 15. Variation of $\eta_2^0$ with shear stress change rule.**

## 5. Conclusions

In this paper, the improved Nishihara model is validated using the results of creep tests on frozen chalk-concrete contact surfaces. The main research results are as follows:

1. It can be obtained from the test data that, in the same time range, the specimen with larger contact surface roughness corresponds to a larger destructive shear stress and a larger long-term strength limit. When the roughness $R$ is 0 mm~1.225 mm, the specimen corresponds to a long term strength of 32.84 kPa~34.57 kPa. At the same roughness, the creep deformation of the contact surface is more significant with the increase of the shear stress $\tau$.

2. The traditional Nishihara model has been improved by transforming its stationary components into non-stationary elements and introducing a damage factor. It can provide a relatively accurate description of the time-dependent creep deformation at the interface between frozen soil and concrete under different roughness conditions, especially during the acceleration phase of creep. The calculated values of the model agree well with the experimental results and have only minimal errors

3. An analysis of the parameters of the improved model reveals the following trends: (i) As the shear stress continues to increase, both the Hooke body shear modulus ($G_0$), and the Kelvin body shear modulus ($G_1$), exhibit a first-order exponential decay. Parameter $\eta_1^0\tau^\beta$ decreases and then increases, while parameter $\eta_2^0$ follows the opposite trend of increasing first and then decreasing. (ii) With an increase in the roughness of the contact interface, $G_0$, $G_1$, and $\eta_2^0$ gradually increase, while parameter $\eta_1^0\tau^\beta$ progressively decreases. (iii) The damage factor ($D$) shows a monotonically increasing trend with time and eventually converges to the numerical value of 1.

## Author Contributions

**Conceptualization:** Fei He.

**Data curation:** Erqing Mao, Qingquan Liu.

**Formal analysis:** Fei He.

**Funding acquisition:** Xu Wang.

**Methodology:** Fei He, Wanyu Lei.

**Project administration:** Xu Wang.

**Resources:** Hangjie Chen.

**Supervision:** Hangjie Chen.

**Validation:** Fei He.

**Visualization:** Qingquan Liu.

**Writing – original draft:** Fei He.

**Writing – review & editing:** Wanyu Lei, Erqing Mao.

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
