## [Decision Letter · Decision Letter 0]

28 Nov 2023

PONE-D-23-35040Creep constitutive modeling of permafrost-concrete interface shear strength considering stress levelPLOS ONE

Dear Dr. He,

Thank you for submitting your manuscript to PLOS ONE. After careful consideration, we feel that it has merit but does not fully meet PLOS ONE’s publication criteria as it currently stands. Therefore, we invite you to submit a revised version of the manuscript that addresses the points raised during the review process.

We look forward to receiving your revised manuscript.

Kind regards,

Mario Milazzo

Academic Editor

PLOS ONE

Journal Requirements:

"The authors appreciate the National Natural Science Foundation of China (Grant No. 41902272) 

and Basic Research Innovation Group Project of Gansu Province, China (Grant No. 21JR7RA347)."

Please remove any funding-related text from the manuscript and let us know how you would like to update your 

Funding Statement. Currently, your Funding Statement reads as follows: 

"The author(s) received no specific funding for this work"

7. We note you have included a table to which you do not refer in the text of your manuscript. Please ensure that you refer to Table 3 in your text; if accepted, production will need this reference to link the reader to the Table.

8. Please upload a copy of Supporting Information Figure/Table/etc. Supporting information which you refer to in your text on page 22-23.

Reviewers' comments:

Reviewer's Responses to Questions

**Comments to the Author**

1. Is the manuscript technically sound, and do the data support the conclusions?

Reviewer #1: Yes

Reviewer #2: Yes

2. Has the statistical analysis been performed appropriately and rigorously? 

Reviewer #1: Yes

Reviewer #2: N/A

3. Have the authors made all data underlying the findings in their manuscript fully available?

Reviewer #1: Yes

Reviewer #2: No

4. Is the manuscript presented in an intelligible fashion and written in standard English?

Reviewer #1: Yes

Reviewer #2: Yes

5. Review Comments to the Author

Reviewer #1: The authors present a compelling article on "Creep constitutive modeling of permafrost-concrete interface shear strength considering stress level." The focus of the study is the creep behavior at the pile-soil interface. Overall, the paper is well-structured and written, introducing a novel perspective on creep constitutive modeling of permafrost-concrete interface shear strength. However, based on the results and methods employed, the reviewer recommends revisions that address specific comments outlined below.

Title:

The title should explicitly convey that the creep tests were conducted under -1°C conditions, a crucial detail absent from the current title.

Abstract:

The abstract requires reorganization to emphasize the primary research findings, adopting a logical structure that briefly covers introduction, problem statement, methodology, results, and conclusion. Including quantitative information in the abstract, such as key results, will enhance the reader's understanding of the study's significance.

Introduction:

The literature review needs completion, and it's unclear what sets this paper apart. Updating the literature review is essential, considering recent works like https://doi.org/10.1007/s12517-021-09394-0. The introduction should better articulate the research's importance and contributions.

Introduction:

Provide additional details about the research gap, emphasizing the significance of your study. Ensure a smooth transition from the introduction to the subsequent sections of the paper.

Method:

The test details lack clarity, and material properties are missing. A more detailed description of the test methods, including material properties, is necessary for comprehensive understanding.

Method:

Include information about the surface concrete used, emphasizing details on roughness. This addition will contribute to a more thorough comprehension of the study.

Results:

Provide additional context for the results by discussing their relevance to the research question or hypothesis. Address any limitations and potential sources of bias or error in the study.

Discussion:

Expand on the broader implications of the findings and their relevance to the field or real-world applications. Strengthen the interpretation of the results to make the paper more engaging.

Conclusion:

Enhance the interpretation of results in the conclusion, supporting it with quantitative findings. Provide a more detailed conclusion that thoroughly summarizes the study's outcomes.

Incorporating these revisions will improve the clarity, coherence, and overall impact of the paper.

Reviewer #2: Good work, minor comments.

- Lots of equation parameters are not defined or introduced, please verify.

- The statistics of related to how well the model fits the experimental data are not presented.

- Table 5 should be modified and reaaranged, the presentation is not adequate.

- Conclusions should be presented using bullet points to clearly highlight each of the findings.

6. PLOS authors have the option to publish the peer review history of their article (what does this mean?). If published, this will include your full peer review and any attached files.

Reviewer #1: **Yes: **Meysam Bayat

Reviewer #2: No

---

## [Author Response · Author response to Decision Letter 0]

4 Jan 2024

Academic editor's requirements:

Response: Based on the downloaded template files, we have revised the article to ensure meet PLOS ONE's style requirements.

2. Please update your submission to use the PLOS LaTeX template.

Response: I'm sorry we're not proficient in using LaTeX for paper layout. But we reformatted the content of the paper according to LaTex's formatting requirements 

3. We suggest you thoroughly copyedit your manuscript for language usage, spelling, and grammar.

Response: We apologize for the language usage, grammar and spelling issues in our paper. We have checked and changed the full text.

4. We note that the grant information you provided in the ‘Funding Information’ and ‘Financial Disclosure’ sections do not match. When you resubmit, please ensure that you provide the correct grant numbers for the awards you received for your study in the ‘Funding Information’ section.

Response: We are very sorry for not matching the grant information in the ‘Funding Information’ and ‘Financial Disclosure’ sections. We identified the grant information as “The research described in this paper was financially supported by the National Natural Science Foundation of China (Grant No. 41902272) and the Basic Research Innovation Group Project of Gansu Province, China (Grant No. 21JR7RA347).” 

5. Thank you for stating the following in the Acknowledgments Section of your manuscript: "The authors appreciate the National Natural Science Foundation of China (Grant No. 41902272) and Basic Research Innovation Group Project of Gansu Province, China (Grant No. 21JR7RA347)." We note that you have provided funding information that is not currently declared in your Funding Statement. However, funding information should not appear in the Acknowledgments section or other areas of your manuscript. We will only publish funding information present in the Funding Statement section of the online submission form. Please remove any funding-related text from the manuscript and let us know how you would like to update your Funding Statement. Currently, your Funding Statement reads as follows: "The author(s) received no specific funding for this work". Please include your amended statements within your cover letter; we will change the online submission form on your behalf.

Response: We are very sorry for adding funding-related text to the manuscript. As required by the journal, we have removed funding-related text in the manuscript. In addition, we identified the Funding Statement as “National Natural Science Foundation of China (Grant No. 41902272) and the Basic Research Innovation Group Project of Gansu Province, China (Grant No. 21JR7RA347).” 

6. In your Data Availability statement, you have not specified where the minimal data set underlying the results described in your manuscript can be found. PLOS defines a study's minimal data set as the underlying data used to reach the conclusions drawn in the manuscript and any additional data required to replicate the reported study findings in their entirety. All PLOS journals require that the minimal data set be made fully available.

Response: We apologize that we did not specify where to find the minimal dataset on which our experimental results are based. We have uploaded the data into an online database, which can be found at the link below. 

https://doi.org/10.6084/m9.figshare.24935673.v2

7. We note you have included a table to which you do not refer in the text of your manuscript. Please ensure that you refer to Table 3 in your text; if accepted, production will need this reference to link the reader to the Table.

Response: We are very sorry that we have omitted the description of Table 3 in the text. We added a description concerning Table 3 in the text.

8. Please upload a copy of Supporting Information Figure/Table/etc. Supporting information which you refer to in your text on page 22-23.

Response: We are very sorry that we have not uploaded supporting information. We have uploaded a copy of Supporting Information in our text on page 17.

 

Reviewer #1:

1. Comment: Title: The title should explicitly convey that the creep tests were conducted under -1°C conditions, a crucial detail absent from the current title. We understand that the terms of hazard, vulnerability and risk are different. That is important clarify this concern by the authors. Please check all text and abstract for this concern.

Response: As suggested by the reviewer, the test conditions for creep testing should be indicated in the title. We added the -1°C condition to the title in the paper. We understand the difference of hazard, vulnerability and risk. Hazard is a potential adverse condition, vulnerability is the sensitivity of a system or individual to a hazard, and risk is the result of the interaction of hazard and vulnerability, combining likelihood and potential damage. We have changed the contents of the abstract and text to avoid this problem.

2. Comment: Abstract: The abstract requires reorganization to emphasize the primary research findings, adopting a logical structure that briefly covers introduction, problem statement, methodology, results, and conclusion. Including quantitative information in the abstract, such as key results, will enhance the reader's understanding of the study's significance.

Response: As suggested by the reviewer, the abstract should briefly cover the content of the research and include quantitative information on key findings to enhance the reader's understanding of the significance of the study. Therefore, the abstract section of the study was rewritten and quantitative information on the main results was added to enhance the reader's understanding of the research.

3. Comment: Introduction: The literature review needs completion, and it's unclear what sets this paper apart. Updating the literature review is essential, considering recent works like https://doi.org/10.1007/s12517-021-09394-0. The introduction should better articulate the research's importance and contributions.

Response: As suggested by the reviewer, the introduction should better articulate the research's importance and contributions. Therefore, we completed the literature review and set forth the difference of this paper, recent works were added in line 71-93 of this paper.

4. Comment: Introduction: Provide additional details about the research gap, emphasizing the significance of your study. Ensure a smooth transition from the introduction to the subsequent sections of the paper.

Response: As suggested by the reviewer, the details about the research gap and significance of study are necessary, so we have added the detail differences of research to ensure a smooth transition from the introduction to the subsequent sections of the paper The addition is in line 94~117 of this paper.

5. Comment: Method: The test details lack clarity, and material properties are missing. A more detailed description of the test methods, including material properties, is necessary for comprehensive understanding.

Response: As suggested by the reviewer, we have added experimental apparatus, test methods and material properties in lines 234~304.

6. Comment: Results: Provide additional context for the results by discussing their relevance to the research question or hypothesis. Address any limitations and potential sources of bias or error in the study.

Response: As suggested by the reviewer, we have added to more information about two hypotheses in the text and explained the limitations of the study and the sources of the bias at the results. Added in line 140-145, 150-155, 353-360

7. Comment: Discussion: Expand on the broader implications of the findings and their relevance to the field or real-world applications. Strengthen the interpretation of the results to make the paper more engaging.

Response: As suggested by the reviewer, we have illustrated the linkage of this study to practical engineering and explain how the results are complementary to the current study. In line 397-414 of the text.

Comment: Conclusion: Enhance the interpretation of results in the conclusion, supporting it with quantitative findings. Provide a more detailed conclusion that thoroughly summarizes the study's outcomes. Incorporating these revisions will improve the clarity, coherence, and overall impact of the paper.

Response: As suggested by the reviewer, quantitative and comprehensive conclusions are important to the clarity and coherence of the paper. Thus, we add detailed conclusions and quantitatively describe the main findings to improve the clarity, coherence, and overall impact of the paper.

 

Reviewer #2:

1. Comment: Lots of equation parameters are not defined or introduced, please verify.

Response: As suggested by the reviewer, we have added some definitions and descriptions of equation parameters.

2. Comment: The statistics of related to how well the model fits the experimental data are not presented.

Response: As suggested by the reviewer, the statistics are very important .We listed the statistics for the fitted data in Table 1

3. Comment: Table 5 should be modified and reaaranged, the presentation is not adequate.

Response: As suggested by the reviewer, we have reorganized table 5 to make it clearer and more adequate.

4. Comment: Conclusions should be presented using bullet points to clearly highlight each of the findings.

5. Response: As suggested by the reviewer, conclusions should be presented using bullet points to clearly highlight each of the findings. We presented the findings of the paper in points, highlighting each research finding.

---

## [Decision Letter · Decision Letter 1]

15 Jan 2024

Creep constitutive modeling of the shear strength of the permafrost-concrete interface considering the stress level at -1℃

PONE-D-23-35040R1

Dear Dr. He,

We’re pleased to inform you that your manuscript has been judged scientifically suitable for publication and will be formally accepted for publication once it meets all outstanding technical requirements.

Kind regards,

Mario Milazzo

Academic Editor

PLOS ONE

Reviewers' comments:

Reviewer's Responses to Questions

**Comments to the Author**

1. If the authors have adequately addressed your comments raised in a previous round of review and you feel that this manuscript is now acceptable for publication, you may indicate that here to bypass the “Comments to the Author” section, enter your conflict of interest statement in the “Confidential to Editor” section, and submit your "Accept" recommendation.

Reviewer #1: All comments have been addressed

Reviewer #2: All comments have been addressed

2. Is the manuscript technically sound, and do the data support the conclusions?

Reviewer #1: Yes

Reviewer #2: Yes

3. Has the statistical analysis been performed appropriately and rigorously? 

Reviewer #1: Yes

Reviewer #2: Yes

4. Have the authors made all data underlying the findings in their manuscript fully available?

Reviewer #1: Yes

Reviewer #2: Yes

5. Is the manuscript presented in an intelligible fashion and written in standard English?

Reviewer #1: Yes

Reviewer #2: Yes

6. Review Comments to the Author

Reviewer #1: The author addressed all the comments raised, so I have no objections to the paper and recommend it for priority publishing.

Reviewer #2: Thank you for reviewing the paper. After reviewing the revised version, I think it is acceptable for publication.

7. PLOS authors have the option to publish the peer review history of their article (what does this mean?). If published, this will include your full peer review and any attached files.

Reviewer #1: **Yes: **Meysam Bayat

Reviewer #2: No

---

## [Editor Report · Acceptance letter]

8 Apr 2024

PONE-D-23-35040R1 

PLOS ONE

Dear Dr. He, 

I'm pleased to inform you that your manuscript has been deemed suitable for publication in PLOS ONE. Congratulations! Your manuscript is now being handed over to our production team.

Kind regards, 

on behalf of

Dr. Mario Milazzo 

Academic Editor

PLOS ONE